# An Effective and Efficient Generation Framework for Condensing the Graph Repository

## Abstract

Graph repositories with multiple graphs are increasingly prevalent in various applications. As the amount of data increases, training neural networks on graph repositories becomes increasingly burdensome. However, existing condensation methods focus more on reducing the size of a single graph. They fail to address the challenges of efficiently and effectively compressing multiple data graphs. In this work, we propose a novel end-to-end Graph Repository Condensation (GRCOND) framework that effectively condenses a large-scale graph repository with multiple graphs, while preserving task-relevant structural and feature information. Unlike traditional methods, our approach pretrains a dataset-specific GNN model to create and optimize synthetic graphs, enabling us to capture both intra-graph structures and inter-graph relationships, and thus providing a more holistic representation of the repository. Through experiments, our proposed approach achieves higher accuracy and retains features across different compression ratios, highlighting the potential of our framework to accelerate GNN training and expand the applicability of graph-based machine learning in resource-constrained environments.

## 1 Introduction

Graph repositories have become fundamental to modern data analysis across diverse domains (Gilmer et al., 2017). From temporal network evolution analysis (Liu et al., 2021) and biological interaction studies (Fout et al., 2017; Huang et al., 2023) to personalized recommendation systems (Fan et al., 2019; Ying et al., 2018), such repositories encode complex relational patterns critical for advancing scientific and industrial applications. As repositories grow in scale and complexity, training Graph Neural Networks (GNNs) on these datasets becomes computationally prohibitive (Hamilton et al., 2017), hindering rapid experimentation and deployment in the resource-constrained settings. As a result, many dataset condensation methods have emerged, such as trajectory matching (Jin et al., 2022a), distribution matching (Zhao & Bilen, 2023), and kernel-based distillation (Xu et al., 2023). Due to the advancement of neural networks and the simplicity of image datasets, these methods achieve effective condensation.

Although existing condensation methods (Gao et al., 2024; Khoshraftar & An, 2024; Dai et al., 2019) have shown promise in reducing training costs, they predominantly focus on single-graph scenarios. These approaches often fall short in capturing the structural diversity and inter-graph relationships inherent in multi-graph repositories. These datasets introduce unique challenges that are not present in single-graph settings (Tang et al., 2015; Dai et al., 2019), such as the need to preserve intergraph relationships, structural diversity, and scalability as the dataset size grows (Velickovic et al., 2018; Kipf & Welling, 2017; Xu et al., 2018). Therefore, *how to efficiently condense a large graph data repository into an extremely small graph data set is the main focus of our paper*.

To address this problem, we propose a novel graph repository condensation framework (GRCOND) that effectively condenses a large-scale graph repository with multiple graphs. Our method can handle the unique properties of graph repositories with multiple graphs, retaining both intra-graph and inter-graph information, which are vital for tasks such as classification and anomaly detection.

To address the issue of uneven quality in small graphs within the graph repository and the high overhead associated with directly optimizing graph data, we developed a new sample optimization method. We first search for the cluster center for each category in the graph repository, which is the graph with the smallest sum of distances to other samples within the category. After selecting these

representative samples, we use the pre-trained model to obtain the corresponding latent vectors, and then optimize these latent vectors during the training process.

Unlike other condensation methods, we use an end-to-end optimization approach. We trained a network to embed the node features and structural features of the graph together, and trained two decoder networks separately during restoration. This reduces the optimization overhead in the compression process without destroying the correspondence between the two, and also retains their respective characteristics. This paper makes the following four major contributions:

- We establish the first fully end-to-end condensation framework for graph repositories, unifying structural preservation and feature distillation via latent space bi-level optimization.
- We propose a framework for condensing a graph repository with multiple graphs (GRCOND), which preserves both intra-graph and inter-graph information.
- We propose a new optimization strategy in gradient matching, which utilizes a pre-trained model as the optimization tool and employs the latent vectors as optimization targets to address the discreteness problem of graph data, while also establishing an identical distribution relationship between the generated graphs.
- We conduct comprehensive experiments on various repositories and various GNNs to show the effectiveness and versatility of our proposed framework.

## 2 RELATED WORK

**Dataset Condensation**. We are witnessing an increasing number of dataset condensation techniques applied to real datasets. It works by generating a small subset of synthetic data, ensuring that it achieves similar performance to the full repository when training a deep learning model. Zhao et al. (Zhao et al., 2021a) formulate this goal as a gradient matching problem between the gradients of deep neural network weights that are trained on the original and their synthetic data. Jin et al. (Jin et al., 2022b) expand its application to graph-structured data where the samples (nodes) are interdependent. However, their methods do not apply to a graph repository with multiple graphs, which have very strong structural characteristics and strong connections between graphs. In this work, we generalize the problem of dataset condensation to the condensation of a graph repository comprising multiple graphs, and we seek a new approach to jointly learn the synthetic node features and graph structure.

**Graph Sparsification / Coarsening / Condensation**. Graph sparsification (Hashemi et al., 2024) focuses on reducing the number of edges in a graph while preserving key properties, such as pairwise distances (Peleg & Schäffer, 1989), cut values (Karger, 1994), or spectral characteristics, including eigenvalues and eigenvectors (Kipf & Welling, 2017; Spielman & Teng, 2011). In contrast, graph coarsening reduces number of nodes by aggregating original nodes into supernodes while maintaining structural and functional properties of the graph (Loukas, 2019; Loukas & Vandergheynst, 2018; Deng et al., 2020; Xu et al., 2019). This is typically achieved by defining the connections of super-nodes to approximate the behavior of the original graph, enabling efficient analysis and computation on the coarser representation (Sun et al., 2020). Among the many ways to reduce graph data storage, we are more concerned about graph repository condensation. It does not reduce the number of nodes or edges, but reduces the number of graphs, which requires paying attention to the connections between graphs while considering the connections between nodes.

## 3 PROBLEM DEFINITION

Dataset condensation is particularly relevant in scenarios requiring computational efficiency, model adaptability, and privacy-preserving machine learning solutions. It is a machine learning technique that aims at synthesizing a smaller and highly informative version of a repository.

**Definition 1** *(Dataset Condensation) Given a large repository consisting of $|T|$ pairs of a training object and its class label $D_o = \{(x_i, y_i)\}|_{i=1}^{|T|}$ where $x \in X \subseteq R, y \in \{0, ..., C-1\}$, the target is to learn a condensed set $D_S$ which can train the neural network $\phi$ on them, and this $\phi$ can be used directly on Z.*

Dataset condensation methods primarily focus on images or tabular data, aiming to reduce the repository size without sacrificing task performance. In this paper, we will extend these techniques to

graph repositories. It introduces unique challenges due to the intricate structure of graphs, including node relationships, edge connections, and feature distributions, which must be preserved in the condensation process.

**Definition 2** *(Graph, G) A graph G is a fundamental data structure defined as a pair G = (V, E, X), where V is a finite set of nodes, which represent the entities in the system. $E \in V \times V$ is a set of edges that represent the relationships between the entities. $X \in \mathbb{R}^{N \times F}$ is a matrix containing node feature information, where N and F are the number of nodes and the node features, respectively.*

Based on the basic data structure, we will define the core issue of this paper.

**Definition 3** *(**Condensation for Graph Repository with Multiple Graphs**) Given a large set of graphs $D_G = \{(G_1, y_1), (G_2, y_2), ..., (G_N, y_N)\}$, the goal is to condense $D_G$ into a extremely smaller set $D_S(|D_S| \ll |D_G|)$ such that a model trained on $D_S$ performs similarly to a model trained on the full repository $D_G$. Data in $D_S$ are all newly generated.*

By reducing the size and complexity of graph repositories while preserving essential structural, feature, and task-specific information, condensation for a graph repository with multiple graphs enables scalable and effective downstream applications in diverse domains such as chemistry, biology, and social networks.

## 4 CONDENSATION FOR GRAPH REPOSITORY

In this section, we introduce our proposed method for graph set condensation in detail. First, we present our overview framework. Then, we explain in turn the implementation details of each module, including explaining how to initialize model parameters, synthesize repositories, and match two repositories through the training trajectory of the same model. After that, we will demonstrate our unique approach to optimization.

### 4.1 OVERVIEW

Figure 1 shows the general framework of the condensation for the graph set. This figure consists of two parts, where the left part represents the initialization and optimization of our condensed graph repository, and the right part represents the training phase. We first use an arbitrary pretrained GNN model to learn the characteristics of the structural information of the graphs. In all our algorithm processes, the network parameters of this pretrained model will not be updated, and only its learning of structural information and generation functions will be used.

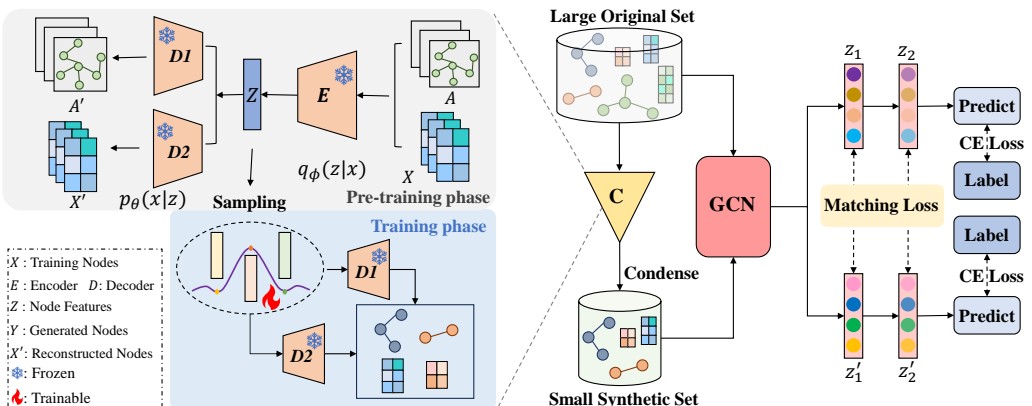

Figure 1: Framework of Condensation for Graph Repository

During the initialization of network parameters and synthetic repositories, we utilize the embedding part of the pre-trained model to extract the embedding matrix, which captures the representations of the graph data. Then sample the embedding vector to obtain the initial embedding of the synthetic data set. In the training phase, we employ gradient matching to align the training trajectories of the two repositories and utilize the matching loss to update the embedding vectors. Then we decode these vectors to restore the structure and feature information of the synthetic repository.

## 4.2 PRETRAINING

We utilize GCN to obtain the embedding of each small graph, thereby leveraging the generalization capabilities of pre-trained GNNs to capture the structural semantics of graphs. During our formal training process, the pretrained GNN parameters are frozen and only used as a feature extractor to prevent multi-objective optimization. In the restoration stage, we pre-trained two models to decode structural features and node features, respectively, because structural features typically follow a sparse discrete distribution, and decoding requires emphasizing the locality and sparsity of topological relationships. Node features are mostly continuous, and decoding must retain semantic associations and maintain smoothness. The update process of the synthetic dataset after using the model can be expressed as follows:

$$S'_{t+1} = D_\psi(Z_{t+1}) = D_\psi\left(Z_t - \eta\, J_{D_\psi}(Z_t)^\top \nabla_S \mathcal{J}(S_t)\right), \tag{1}$$

where $S'$ is the synthetic graph generated by the decoder and $Z_t$ is the current latent variable. $D_\psi$ is a fixed decoder. $J_{D_\psi}(Z_t)$ is the Jacobian matrix of the decoder at $Z_t$ and $\eta$ is the learning rate. We then use the first-order Taylor expansion approximation:

$$S'_{t+1} \approx D_\psi(Z_t) - \eta\, J_{D_\psi}(Z_t)\, J_{D_\psi}(Z_t)^\top \nabla_S \mathcal{J}(S_t) \approx S_t - \eta\left[J_{D_\psi}(Z_t) J_{D_\psi}(Z_t)^\top\right] \nabla_S \mathcal{J}(S_t), \tag{2}$$

where $\nabla_S \mathcal{J}(S_t)$ is the gradient of the objective function with respect to the synthetic dataset. Since the decoder remains fixed throughout the process, our approach establishes a deterministic mapping relationship. Even when employing an indirectly trained decoder, it can achieve similar results to conventional synthetic dataset optimization methods.

## 4.3 INITIALIZATION PHASE

We sample embedding vectors from the original graphs to generate the initial embedding of the synthetic graph, and then decode them into the adjacency matrix and node features of the synthetic graph using the pre-trained model. Due to the uneven distribution of small graph quality, random sampling will result in an unstable quality of the synthetic graph. So we select cluster centers for the embedding vectors to select the most representative subgraph:

$$cl_i^{(t)} = \arg\min_k \left\|\mathbf{z}_i - \mathbf{m}_k^{(t)}\right\|_2^2, \ \forall i \in \mathcal{I}_c, \quad \mathbf{m}_k^{(t+1)} = \frac{1}{\left|\mathcal{S}_k^{(t)}\right|} \sum_{i \in \mathcal{S}_k^{(t)}} \mathbf{z}_i, \ \mathcal{S}_k^{(t)} = \left\{i \mid cl_i^{(t)} = k\right\}, \tag{3}$$

where $m_k^{(t)}$ is the centroid vector of the t round and k is the cluster ID. $z_i$ is our embedding vectors. $cl_i^{(t)}$ is the cluster ID of the i-th vector in round t. We first assign each embedding vector to the closest cluster based on the distance, and then obtain the new cluster center by averaging the vectors within each cluster. This choice can ensure that the initialization embedding vector has high quality.

Then, we determine a neural network model $GNN_{\theta_0}$ for training, where $GNN_\theta$ denotes the GNN model parameterized with $\theta$. And $\theta_0$ is randomly sampled from a specific distribution $P_\theta$ to make the trained synthetic repository independent of the network's initialization parameters. After that, we can preliminarily represent our target formula. Our target is to learn an extremely small synthetic graph repository $D_S$ such that a GNN trained on $D_S$ can achieve great performance comparable to that of a GNN trained on the much larger original repository $D_O$. Thus, the objective can be formulated as the following bi-level problem:

$$D_S = \min_{D_S} E_{\theta_0 \sim P_\theta}[L(GNN_{\theta_S}(D_G), Y_G)] \ \ s.t. \ \theta_S = \arg\min_\theta\ L(GNN_{\theta(\theta_0)}(D_S), Y_S), \tag{4}$$

where $GNN_\theta$ denotes the GNN model parameterized by $\theta$, $\theta_S$ denotes the parameters of the model trained on $D_S$, and L denotes the loss function used to measure the difference between the model's predictions and the ground truth.

## 4.4 TRAINING PHASE

To solve the problem in equation 4, we need to make the networks trained on the two repositories closer. The goal is to find a small synthetic repository that best represents the information in the

original repository to make the network parameters trained by the two repositories to be similar, which can be expressed as follows:

$$D_S = \min_{D_S} \sum_{t=0}^{T-1} Dis(\theta_S^t, \theta_O^t) \; with \tag{5}$$

$$\theta_S^{t+1} = \Delta_\theta(L(GNN_{\theta_S^t}(D_S), Y_S)) \; and \; \theta_O^{t+1} = \Delta_\theta(L(GNN_{\theta_O^t}(D_O), Y_O)),$$

so that the network trained by the synthetic graph repository can also have similar effects on the large repository. However, the overhead of directly matching network parameters is very high, so we choose to match the paths of training the networks of the two repositories, that is, to match the gradients that descend during the training process.

Our approach does not involve solving a nested loop optimization and unrolling the entire training trajectory of the inner problem, which can be prohibitively expensive. Instead, we follow the gradient matching method proposed in (Zhao et al., 2021b), which aims to match the network parameters between large-real and small-synthetic training data by matching their gradients at each training step:

$$\nabla_\theta L(GNN_{\theta^t}(D), Y) = (\theta^{t+1} - \theta^t)/\mu, \tag{6}$$

where $\nabla$ is the gradient of the network's descent at the corresponding step, and $\mu$ is the learning rate. In this way, the training trajectory on the small synthetic graph repository $D_S$ can mimic that on the large real graph repository $D_O$. The gradients matching process for GNN can be modeled as follows:

$$\min_{D_S} \sum_{t=0}^{T-1} Dis(\nabla_{D_S}^t, \nabla_{D_O}^t), \tag{7}$$

where $\nabla_{D_S}^t$ represents the gradient of the repository $D_S$ at the t step of the network, and $Dis$ is a function that calculates the distance between two gradients. To more accurately match the training gradients between the synthetic dataset and the original dataset, we performed a gradient matching operation on each class, allowing the synthetic repository to learn the differences between classes. Additionally, as demonstrated in the work on reconstructing data from gradients, large batch sizes tend to make reconstruction more challenging. We sample a fixed-size set of neighbors on the original graph in each training round, employing a mini-batch training strategy.

This phase helps preserve the inter-graph information. We use gradient matching to optimize embedding parameters, ensuring that synthetic data produces model update directions equivalent to those of the original data in downstream task training. For example, during the training of a graph classification task, the embedding vectors corresponding to graphs of the same category are adjusted to conform to a similar distribution, while the distance between graphs of different categories is increased. In this way, the synthetic dataset can learn the relationship between graphs.

## 4.5 OPTIMIZATION OF CONDENSED GRAPH REPOSITORY

We calculate the distance between the gradients of the two repositories obtained in the model training as the loss value. Then, the goal of optimization needs to be considered. For repositories with multiple graphs, we use our pretrained decoders to optimize synthetic repositories. Since repositories with multiple graphs generally conform to a specific distribution and exhibit distinct characteristics in their graph structure, such as those found in molecular, biological, and chem-informatics repositories. The pre-trained GNN model $\phi_{gen}$ will learn the distribution of the graph structures in the repository, meaning that the graphs generated by the model essentially conform to the distribution of the corresponding repository. The resulting graph will contain more information about the original repository and facilitate our optimization work. In our work, the optimization target is the latent vector in the graph generation model, specifically a set of latent vectors sampled during the initialization phase. The purpose is to transform the discrete adjacency matrix into a continuous embedding, which can be expressed as follows:

$$Z^{t+1} = Z^t - \nabla_Z Dis(\nabla_{GNN_{D_S}}^t, \nabla_{GNN_{D_O}}^t), \tag{8}$$

where $Z^t$ is the hidden vector at the t-th round of training, and $\nabla_Z$ is the gradient calculated for each position of Z using the loss value generated by gradient matching to optimize the hidden vector. We

can achieve better results by performing gradient descent on the loss value on continuous data. Then, the latent vector after gradient descent is converted into an adjacency matrix using the decoding part of the trained graph generation model.

$$
\begin{aligned}
A_g &= \phi_{gen}(Z_g), \\
A_g^{(k)}{}_{(i,j)} &= \psi\left(\alpha \cdot f^{(k)}(Z_g, i) + \beta \cdot f^{(k)}(Z_g, j)\right), \\
f^{(k)}(Z_g, i) &= \sigma\left(W^{(k)} Z_g^{(k-1)}{}_i\right),
\end{aligned}
\tag{9}
$$

where $A_g$ is the adjacency matrix corresponding to the $g^{th}$ graph. $\sigma$ and $\psi$ are the activation function of pretrained GNN model and $W$ is pretrained parameters. Since the decoder part of the graph generation model $\phi_{gen}$ has been trained to generate graphs that conform to the distribution, its parameters will not change during the whole process to ensure that the reconstructed graph also contains the intra-graph information learned from the pretraining phase.

### 4.6 Implementation

Algorithm 1 shows the overall process of condensation for the graph repository. In lines 1-2, the original dataset and pretrained model are given. Line 4 is the initialization phase of the synthetic graph, where we sample embedding vectors by Equation equation 3. Lines 7-8 calculate the loss value of the two datasets on the downstream task. Line 9 calculates the difference between the descent gradient of the synthetic dataset and the original dataset. Lines 10-12 optimize the embedding vector based on the loss value and utilize the pre-trained decoder to update the synthetic dataset. Line 13 uses the loss value of the original dataset to optimize the model for downstream tasks.

---

**Algorithm 1:** Our Condensation Algorithm

1 **Input:** Training graph set $D_G = \{G_1, G_2, ..., G_N\}$, number of outer-loop steps $K$, randomly initialized weights $P_{\theta_0}$, number of inner-loop steps $T$, number of classes $C$, GNN $\phi_\theta$, loss function $\ell$ for the graph classification, pre-trained GNN model $\phi_{gen}$.

2 **Output:** Condensed graph set $D_S = \{G_1, G_2, ..., G_M\}$, where $M \ll N$ and $D_S \not\subset D_G$.

3 **for** $k = 0, \cdots, K-1$ **do**

4    sample $\theta_0 \sim P_{\theta_0}$ and $Z_{D_S} \sim P_Z$;

5    **for** $t = 0, \cdots, T-1$ **do**

6      **for** $c = 0, \cdots, C-1$ **do**

7        Sample $B_c^{D_G} \sim D_G$;

8        Compute $L_{D_G} = \ell(\phi_{\theta_t}(B_c^{D_G}), Y)$, $L_{D_S} = \ell(\phi_{\theta_t}(A_{D_S}, X_{D_S}), Y)$;

9        Compute $L_g \leftarrow D(\nabla L_{D_G}, \nabla L_{D_S})$;

10        Update $Z_{D_S} \leftarrow Z_{D_S} - \eta \cdot \nabla L_g(Z_{D_S})$;

11      **end**

12      Update $A_{D_S}, X_{D_S} \leftarrow \phi_{gen}(Z_{D_S})$;

13      $\theta_{t+1} \leftarrow opt_\theta(L_{D_G})$;

14    **end**

15 **end**

16 **return** $D_S$

---

## 5 Experiment

To evaluate the effectiveness of our approach, we implemented our framework on top of Pytorch (version 2.5.0). All experiments were carried out on a workstation with an Ubuntu operating system, an Intel i9-12900K CPU, 128GB of memory, and a NVIDIA GeForce GTX4090 GPU. In this section, we designed comprehensive experiments to answer the following three research questions (RQs).

**RQ1 (Superiority)** What are the advantages of GRCOND compared with state-of-the-art methods?

**RQ2 (Effectiveness)** Can our method effectively condense the repository so that the compressed data set has a similar effect to the original repository?

**RQ3 (Module necessity)** Does each of our modules play its own role and promote the results?

**RQ4 (Meaningfulness)** Can our condensed graphs show the original graph repository's properties?

### 5.1 Experimental Settings

**Datasets**. In this paper, we selected five real-world graph classification datasets, including NCI1, DD from TUDataset (Morris et al., 2020), and ogbg-molhiv, ogbg-molbbbp, and ogbg-molbace from the Open Graph Benchmarks (Hu et al., 2020). Table 1 presents the detailed statistics of the dataset. For these datasets, 80/10/10% of the graphs are randomly split into training/validation/test sets.

**Baselines**. To comprehensively evaluate the performance of our condensation method for the graph repository using the multiple graphs approach, we compared it against a diverse set of baselines,

Table 1: Statistics of the tested real-world graph repositories.

| Dataset | Type | # of Graphs | # of Avg. Nodes | # of Avg. Edges | # of Attributes | # of Classes |
|---|---|---|---|---|---|---|
| NCI1 | Chemical | 4110 | 110 | 64 | 37 | 2 |
| DD | Bioinformatics | 1179 | 284 | 14322 | 89 | 2 |
| ogbg-molhiv | Bioinformatics | 41127 | 25.5 | 27.5 | 9 | 2 |
| ogbg-molbbbp | Bioinformatics | 2,039 | 24 | 26 | 9 | 2 |
| ogbg-molbace | Bioinformatics | 1,513 | 34.1 | 36.9 | 9 | 2 |

including state-of-the-art approaches in graph dataset condensation and sampling. The baselines are categorized as follows. For the condensation methods, we selected representative methods from each classic compression type. DosCond (Jin et al., 2022a) is a graph dataset condensation method based on gradient matching. KiDD (Xu et al., 2023) is based on kernel ridge regression, which utilizes the graph neural tangent kernel instead of optimizing the neural network; however, its computational overhead can be very high for large datasets. Mirage (Gupta et al., 2024) extracts frequent computational tree patterns, thereby reducing the size of the training data while maintaining model performance. However, for non-message passing architectures, Mirage may not be applicable or perform poorly. Meanwhile, we also selected some simple baselines, such as full data training, random subsampling, and k-center. Full data training means training GNNs on the original multi-graph dataset without any condensation. This serves as an upper bound for task performance. Random subsampling involves randomly selecting a subset of graphs or nodes within graphs to create a condensed repository. K-Center selects k center points from the given sample set so that their distances to all samples are minimized. Although naive, this baseline highlights value of task-informed condensation methods.

**Hyperparameter Settings**. We set the hyperparameters for the proposed condensation method on a graph repository using the multiple graphs approach and the baseline approaches. The key hyperparameter settings used in our experiments are summarized below. For the condensation method, the number of condensed graphs was set to 1, 5, 10, 20, and 50 per class. The learning rates for structure and feature are set to 0.001 and 0.0001, respectively, and the Adam optimizer with a weight decay of $5 \times 10^{-4}$ was used. In the evaluation stage, we train the same network for 1,000 epochs on the condensed graph with a learning rate of 0.001.

## 5.2 COMPARISON WITH STATE-OF-THE-ARTS (RQ1)

To demonstrate the superiority of our method, we conducted comprehensive comparative experiments on graph classification tasks to evaluate the performance of our proposed method against other graph repository condensation approaches and sampling. Table 2 presents the detailed comparison results. Column 1 lists the names of five widely used repositories for graph classification, ensuring a diverse range of benchmarks. Column 2 provides the condensation rate for each repository, representing the proportion of the original graph data retained after applying the respective condensation method. Columns 3-8 show the test accuracy achieved using four graph condensation techniques, including our proposed method, under identical condensation rates. Column 9 reports the test accuracy of the original repositories without any condensation, serving as an upper-bound reference for performance.

From this Table, we can see that GRCOND not only condenses the repository effectively but also has the lowest information loss rate. It can achieve a recurrence rate of at least 81.67% in accuracy, even when there is only one graph in each class, and achieve a recurrence rate of up to 98.27% when there are 50 graphs per class. The results demonstrate that, under the same compression rate, the repositories condensed using our method consistently achieve superior test accuracy compared to other methods. This highlights the superiority of our approach in preserving critical information for downstream tasks. Specifically, our method outperforms sampling and repository condensation significantly. The superior performance of our method can be attributed to its ability to jointly preserve structural, feature, and task-specific information across multiple graphs. Unlike traditional methods that focus solely on sparsity or coarsening, our approach optimally condenses the relevant information for downstream tasks, leading to better generalization and reduced computational overhead.

## 5.3 TEST WITH DIFFERENT GNNS (RQ2)

To evaluate the effectiveness of the proposed condensation method for the graph repository with multiple graphs, we performed experiments using various GNN architectures. These experiments were designed to assess whether the condensed graphs generated by our method are compatible

Table 2: Comparison of different baselines and GRCOND on various repositories with attributes. The best performance is **highlighted** in bold.

| Dataset | GPC | Methods | | | | | | Whole Dataset |
|---|---|---|---|---|---|---|---|---|
| | | Random | K-Center | DosCond | KiDD | Mirage | GRCOND | |
| NCI1 (ACC) | 1 | 50.90±2.10 | 51.90±1.60 | 49.20±1.10 | 60.40±0.50 | 50.80±2.20 | **60.64±2.56** | |
| | 5 | 52.10±1.00 | 47.00±1.10 | 51.10±0.80 | 63.20±0.20 | 51.30±1.10 | **64.54±1.74** | |
| | 10 | 55.60±1.90 | 49.40±1.80 | 50.30±1.30 | 64.20±0.10 | 51.70±1.40 | **64.90±1.56** | 80.0±1.8 |
| | 20 | 58.70±1.40 | 55.20±1.60 | 50.30±1.30 | 60.90±0.70 | 52.10±2.20 | **65.53±2.46** | |
| | 50 | 61.10±1.20 | 62.70±1.50 | 50.30±1.30 | 65.40±0.60 | 52.40±2.70 | **69.09±1.16** | |
| DD (ACC) | 1 | 49.70±11.30 | 58.80±6.10 | 46.30±8.50 | 71.30±1.50 | **74.00±0.40** | 69.88±0.84 | |
| | 5 | 40.80±4.30 | 51.30±5.30 | 57.50±5.60 | 70.90±1.10 | - | **71.28±0.64** | |
| | 10 | 63.10±5.20 | 53.40±3.10 | 46.30±8.50 | 71.50±0.50 | - | **72.49±1.56** | 76.9±2.2 |
| | 20 | 56.40±4.30 | 58.50±5.70 | 40.70±0.00 | 71.20±0.90 | - | **71.33±1.92** | |
| | 50 | 58.90±6.30 | 62.30±2.50 | 44.00±6.70 | 71.80±1.00 | - | **73.27±3.24** | |
| ogbg-molhiv (ROC-AUC) | 1 | 0.366±.087 | 0.462±.072 | 0.674±.131 | 0.664±.016 | **0.710±.009** | 0.644±.007 | |
| | 5 | 0.501±.051 | 0.519±.096 | 0.369±.175 | 0.657±.005 | 0.703±.012 | **0.715±.015** | |
| | 10 | 0.554±.031 | 0.471±.054 | 0.457±.214 | 0.632±.000 | 0.513±.055 | **0.646±.009** | 0.701±.028 |
| | 20 | 0.621±.022 | 0.627±.050 | 0.281±.007 | 0.648±.025 | 0.633±.048 | **0.669±.012** | |
| | 50 | 0.625±.062 | 0.680±.049 | 0.455±.214 | 0.587±.038 | 0.588±.067 | **0.688±.014** | |
| ogbg_molbace (ROC-AUC) | 1 | 0.468±.045 | 0.486±.035 | 0.512±.092 | 0.706±.000 | 0.590±.004 | **0.710±.041** | |
| | 5 | 0.312±.019 | 0.553±.024 | 0.555±.079 | 0.562±.000 | 0.419±.010 | **0.671±.035** | |
| | 10 | 0.442±.028 | 0.594±.019 | 0.536±.072 | 0.594±.000 | 0.419±.010 | **0.674±.028** | 0.763±.020 |
| | 20 | 0.510±.023 | 0.512±.031 | 0.484±.080 | 0.640±.011 | 0.423±.011 | **0.643±.036** | |
| | 50 | 0.486±.020 | 0.595±.026 | 0.503±.084 | 0.723±.011 | - | **0.681±.024** | |
| ogbg_molbbbp (ROC-AUC) | 1 | 0.510±.013 | 0.532±.015 | 0.546±.026 | 0.616±.000 | 0.592±.004 | **0.627±.043** | |
| | 5 | 0.522±.014 | 0.581±.022 | 0.519±.041 | 0.607±.005 | 0.431±.013 | **0.620±.033** | |
| | 10 | 0.508±.018 | 0.619±.027 | 0.505±.028 | **0.663±.000** | 0.465±.036 | 0.656±.029 | 0.635±.017 |
| | 20 | 0.567±.010 | 0.546±.012 | 0.493±.031 | 0.677±.001 | 0.610±.022 | **0.680±.015** | |
| | 50 | 0.595±.014 | 0.594±.016 | 0.509±.015 | **0.684±.009** | 0.590±.031 | 0.678±.024 | |

with different GNN models and can maintain high performance across a range of architectures. Additionally, we aimed to examine the transferability of task-specific information preserved in the condensed graphs by testing their performance on unseen GNN architectures. Row 3-7 in Table 3 summarizes the test accuracy results for synthetic repositories trained with one GNN and tested with different networks. The first column lists the GNN models used for training on the condensed repositories (e.g., GCN, GAT, and GraphSAGE), while Columns 2-6 present the test accuracy results for other GNNs when applied to the same task on the condensed repositories. The results demonstrate that condensed repositories consistently deliver high performance across a range of test networks, underscoring the compatibility of our method with diverse GNN architectures. To further highlight the impact of condensation, Row 2 in Table 3 compares the accuracy of each GNN on the complete uncondensed repository. Our condensation method retains information in the original graph repository by comparing the test accuracy on the original repository with that on the condensed repository.

From Table 3, we can see that the repository condensed by DGCNN can be effectively used to train the remaining networks, achieving a test accuracy restoration effect of at least 92.80%. A repository condensed by other methods can also achieve an 87.73% restoration effect in training neural networks. Furthermore, the results reveal that regardless of the GNN used for condensation, the test accuracy of a given network differs by up to 4.33%. This highlights the robustness and effectiveness of the condensed graphs. By preserving task-relevant information, synthetic graphs facilitate effective training and testing across a diverse set of GNN models, making them a valuable tool for reducing repository size without sacrificing performance.

Table 3: Cross-architecture performance in accuracy (%) for condensed 5 graphs/class (with a condensation rate of 1%) in PROTEINS repository

| Test\Train | DGCNN | GIN | GAT | GraphSAGE | GCN |
|---|---|---|---|---|---|
| Full Test | 74.10±0.57 | 66.07±0.92 | 65.17±0.63 | 66.96±0.78 | 61.60±0.84 |
| DGCNN | 71.61±0.73 | 62.50±0.41 | 61.42±1.63 | 62.14±1.18 | 60.72±1.29 |
| GIN | 68.07±0.76 | 59.52±1.06 | 60.73±0.98 | 59.52±0.85 | 58.54±1.57 |
| GAT | 72.03±0.69 | 58.17±0.91 | 60.71±1.67 | 60.39±0.53 | 59.46±0.82 |
| GraphSAGE | 69.17±0.90 | 61.30±0.78 | 61.01±0.81 | 62.04±2.14 | 58.92±2.05 |
| GCN | 69.04±1.40 | 60.82±1.35 | 59.22±1.29 | 58.75±0.79 | 59.64±0.94 |

## 5.4 ABLATION STUDY (RQ3)

To demonstrate the necessity of GRCOND in each main module, we tested various variants of GRCOND to conduct ablation experiments. The results are presented in Table 4, which describes the experiments designed for two modules. For the pretrained model, we designed two variants: one variant excluded the complete pre-trained model. We directly perform gradient descent on the node features and structural features of the graph to optimize graph data. The other variant used the

untrained model to test, demonstrating that even if the pre-trained model is average, it can achieve results similar to those of conventional synthetic dataset optimization methods.

For our optimization module, we designed three specific variants: one variant excludes the optimization operation for the adjacency matrix, another removes the optimization operation for node features, and the third eliminates both optimization components. In the second column of the table, we use three different evaluation indicators to describe the test results specifically. The second row enumerates these variants and GRCOND. Rows 3 and 5 are the accuracy (ACC) of the prediction results. Rows 4 and 6 are the Jaccard scores (JAC) of the predicted results, which is a metric that measures the similarity between two sets, particularly useful for multi-label classification and segmentation tasks. Rows 5 and 7 are the macro f1-scores (MF1), which are an aggregate metric that considers precision and recall across all classes, treating each class equally regardless of its size. From Table 4, GRCOND consistently outperforms its variant models, achieving up to a 16.54% gain in accuracy and a 22.33% improvement in Jaccard score, indicating that GRCOND's operation in the optimization part is reasonable and effective.

Table 4: Performance of the variants on PROTEINS and NCI1

| Dataset | Metric | Variant | | | | | |
|---------|--------|---------|----------------|----------------|-----------|-----------|--------------|
|         |        | w/o VAE | untrained VAE  | pretrained VAE | w/o opt(A)| w/o opt(X)| w/o opt(A&X) |
| PROTEINS | ACC | 67.55±0.71 | 67.07±0.85 | 71.61±0.73 | 68.75±1.73 | 66.85±1.65 | 57.14±2.45 |
|          | JAC | 51.29±1.51 | 51.16±1.82 | 59.61±1.05 | 46.77±1.96 | 44.77±2.13 | 37.28±3.17 |
|          | MF1 | 67.24±1.36 | 67.31±1.15 | 73.40±1.52 | 68.60±1.82 | 66.96±1.55 | 58.46±2.63 |
| NCI1     | ACC | 61.42±1.25 | 61.45±1.72 | 69.09±1.16 | 63.78±1.88 | 60.82±2.21 | 52.55±2.06 |
|          | JAC | 45.83±1.47 | 45.69±0.97 | 51.58±1.02 | 42.96±1.54 | 44.09±3.13 | 35.84±3.47 |
|          | MF1 | 59.01±1.24 | 60.73±1.43 | 68.24±0.89 | 61.31±1.35 | 59.90±2.53 | 53.50±2.50 |

## 5.5 VISUALIZATION (RQ4)

To intuitively demonstrate the meaningfulness of our proposed condensation for the graph repository with multiple graphs method, we provide visualizations comparing the original and condensed graphs. These visualizations utilize colors to represent specific labels. Such visualizations aim to illustrate how our method successfully retains critical structural and feature information even after significant condensation. In Figure 2, we compare a subset of our synthetic repository (on the left) with a corresponding portion of the original repository (on the right). The visual differences between the two classes are

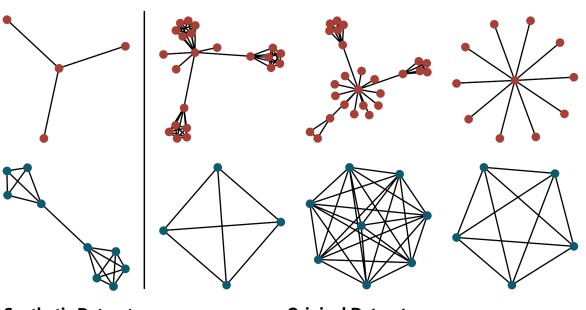

Figure 2: Comparison between synthetic and original datasets

evident in the graphs. For instance, the blue class exhibits a more divergent structure with fewer loops, while the red class demonstrates a tighter configuration with more loops. These distinctions are key to class differentiation and are effectively preserved in the synthetic repository. In particular, our method not only replicates the original graph but also generates a condensed version that retains these crucial structural patterns. This ability to maintain the defining characteristics of the original graphs, while significantly reducing the repository size, underscores the effectiveness of our proposed approach.

## 6 CONCLUSION

In this paper, we present a novel graph condensation framework that effectively condenses large-scale graph datasets into compact synthetic sets while preserving critical structural and semantic information. The proposed algorithm ensures that synthetic graphs not only mimic the statistical properties of the original data but also replicate the training dynamics of graph neural networks (GNNs). The proposed method addresses limitations of conventional dataset condensation techniques, which often fail to handle graph-structured data or rely solely on output-space matching. Experimental validation demonstrates its superiority over random sampling, coreset selection, and graph-level condensation baselines in terms of classification accuracy and structural preservation. Future directions may explore advanced pretrained graph models and extensions to dynamic or heterogeneous graphs.

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

# A    COMSUPTION ANALYSIS

We have conducted some additional experiments to demonstrate the efficiency of our framework. The table below displays the GPU memory usage and the time required for a single training round. It shows that our method is indeed more efficient than other similar condensation tasks. This is because we reduce computational overhead by changing the optimization objective to the latent vector rather than directly performing gradient descent on the graph data.

Table 5: Comparison of running time and GPU memory cost of different methods

| DS | Consumption | DosCond | KiDD | Ours |
|---|---|---|---|---|
| NCI1 | Time(s) | 2.28 | 7.89 | 1.06 |
| | GPU Memory(MB) | 175.62 | 763.93 | 94.15 |
| ogbg-molhiv | Time(s) | 2.53 | 18.72 | 1.01 |
| | GPU Memory(MB) | 389.48 | 748.82 | 289.48 |

We also conducted some additional experiments to demonstrate the effectiveness of our framework. The following table compares the performance of the graphs condensed using our method with that of the original graphs. It shows the GPU memory usage and time required for a single round of training. As shown in the table, our condensed dataset exhibits a significant performance advantage over the original dataset, demonstrating the effectiveness of our method.

Table 6: Comparison of running time and GPU memory cost of original dataset and condensed dataset

| DS | Consumption | Original | Condensed |
|---|---|---|---|
| ogbg-molhiv | Time(s) | 0.6537 | 0.0021 |
| | GPU Memory(MB) | 1489.32 | 28.16 |
| ogbg-molbbbp | Time(s) | 0.5523 | 0.0020 |
| | GPU Memory(MB) | 1264.59 | 27.55 |

Then, we demonstrate the necessity of our method. Our GRCOND is designed for scenarios (e.g., hyperparameter tuning and architecture search that require a variety of GNN tests) involving multiple model training processes using condensed datasets. It is essential to note that in these scenarios, the one-time cost of condensation can be amortized across all downstream training tasks involved, thereby effectively reducing the training time for subsequent tasks. The following shows the training time of an example involving three training tasks on the same dataset ogbg-molhiv, using GCN, GAT, and GIN, respectively. From this table, we can find that the condensation time (i.e., 1287.62s) is indeed larger than the training time for each task using the original ogbg-molhiv dataset. However, when considering all three tasks together, the overall training time using GRCOND (i.e., 1290.60s) is significantly smaller than that of its counterpart (i.e., 2265.50s), demonstrating the superiority of our approach in handling such a scenario.

Table 7: Time consumption in our scenarios

| Scenario | Time (s) |
|---|---|
| Original Data + GCN | 809.70 |
| Original Data + GAT | 765.75 |
| Original Data + GIN | 690.05 |
| Total | 2265.50 |
| Condensation | 1287.62 |
| Condensed Data + GCN | 1.05 |
| Condensed Data + GAT | 0.96 |
| Condensed Data + GIN | 0.97 |
| Total | **1290.60** |

## B  STRUCTURE PRESERVATION TEST

We quantitatively measure information preservation beyond accuracy. We test it through: the average degree of all graphs, the average number of triangles contained, and the average clustering coefficient. It can be seen that the small graphs we generated have a positive effect on preserving the degree of the original dataset.

Table 8: Various structural indicators between the original graphs and the condensed graphs

| DS | | avg. degree | avg. triangle | avg.clustering |
|---|---|---|---|---|
| NCI1 | Original | 4.3088 | 0.0462 | 0.0031 |
| | Condensed | 4.1041 | 0.1754 | 0.0205 |
| PROTEINS | Original | 7.4492 | 27.7438 | 0.5179 |
| | Condensed | 7.2509 | 47.875 | 0.5246 |

We also validated the preservation of inter-graph relationships through multi-faceted evidence. We tested the cosine similarity of the embedding matrices of small graphs within the same class and the cosine similarity of the embedding matrices of different classes. We can observe a class distinction in the embedding matrix. Additionally, we tested the average number of triangles contained in each class of synthetic graphs.

Table 9: Various structural indicators between the graphs inter-class and intra-class

| | cosine_similarity | inter-class cosine_similarity | average triangles for class 0 | average triangles for class 1 |
|---|---|---|---|---|
| PROTEINS | 0.9905 | 0.3245 | 8.2 | 3.4 |
| NCI1 | 0.9864 | 0.4027 | 6.4 | 0.8 |

