# OpenReview forum: "An Effective and Efficient Generation Framework for Condensing the Graph Repository"
_ICLR.cc/2026/Conference — Submitted to ICLR 2026_

### Official Review · Reviewer_pHNi · 2025-10-21

**Soundness:** 2
**Presentation:** 3
**Contribution:** 2
**Rating:** 4
**Confidence:** 4

**Summary:**

This paper introduces GRCOND for condensation of large graph repositories. The method learns a compact set of latent codes that are decoded into synthetic graphs and features. A frozen pretrained encoder provides representational priors, and two dedicated decoders reconstruct topology and attributes. The synthetic set is optimized with class-wise gradient matching so that training on the condensed data yields update trajectories that closely follow those induced by the full corpus. The study evaluates multiple benchmarks across multiple compression ratios, showing consistent improvements over sampling and prior condensation methods, together with notable reductions in training cost.

**Strengths:**

1. The studied problem is important for large-scale graph research and applications, where full training cycles are costly.

2. The document-to-latent pipeline is clear from pretraining through initialization to trajectory matching and decoding.

3. Cost and speed evidence is provided, which strengthens the real-world usefulness of the framework.

**Weaknesses:**

1. Missing important and strong recent baselines. The paper should align budgets and hyperparameter tuning across all methods, add at least one recent distribution matching approach adapted to graphs, and document model selection using a shared validation protocol.

2. Reported averages lack a consistent statement on seeds and confidence intervals. Please provide three to five seeds, report mean and percentile confidence intervals, and run standard significance tests on headline gains.

3. All tasks concern graph classification on molecules and proteins. Results on social or heterogeneous graphs would help test transfer. A small out-of-domain repository would improve external validity.

4. Several symbols are introduced without complete definitions, including $\alpha$, $\beta$, $\psi$, $f$, and $\sigma$. The structure of the distance term Dis($\cdot$) and the batching scheme for class-wise gradient matching are not fully specified. The paper should present the complete loss composition, normalization choices, and batch formation rules in one place.

**Questions:**

1. Are the encoder and both decoders trained strictly on training graphs?

2. What exact distance is used in Dis($\cdot$) and how is it normalized across layers?

3. What is the effect of removing class-wise matching in favor of a global objective?

4. How sensitive is GRCOND to the chosen frozen encoder family?

5. Can you show results on a repository with more classes and significant class imbalance?

---

> ### Author Response · Authors · 2025-11-23
>
> 1.	Response to W1 (baselines)
>
> Thank you for this valuable suggestion. We agree that including more recent and relevant state-of-the-art methods is crucial for a comprehensive evaluation and the reviewer points out the importance of including a distribution matching approach. However, a key challenge we faced is that there is currently no existing distribution matching method specifically designed for graph-level dataset condensation. In response to your feedback, we have conducted new experiments to include the very recent graph-level dataset condensation method, SGDC (Self-supervised Learning for Graph Dataset Condensation) from KDD 2024. We have updated our experimental section and Table 2 in the manuscript to include these comparisons with SGDC, providing a more thorough and up-to-date comparison for graph repository condensation.
>
> |   Dataset  | Gpc |            |            |       ACC      |                |                | Whole Dataset |
> |:----------:|:---:|:----------:|:----------:|:--------------:|----------------|:--------------:|:-------------:|
> |            |     |   DosCond  |    KiDD    |     Mirage     | SGDC           |     GRCOND     |               |
> | NCI1 (ACC) | 1   | 49.20±1.10 | 60.40±0.50 | 50.80±2.20     | **61.26±1.91** | 60.64±2.56     |    80.0±1.8   |
> |            | 5   | 51.10±0.80 | 63.20±0.20 | 51.30±1.10     | 62.32±1.62     | **64.54±1.74** |               |
> |            | 10  | 50.30±1.30 | 64.20±0.10 | 51.70±1.40     | 62.75±1.47     | **64.90±1.56** |               |
> |            | 20  | 50.30±1.30 | 60.90±0.70 | 52.10±2.20     | 62.68±1.73     | **65.53±2.46** |               |
> |            | 50  | 50.30±1.30 | 65.40±0.60 | 52.40±2.70     | 62.79±1.98     | **69.09±1.16** |               |
> |  DD (ACC)  | 1   | 46.30±8.50 | 71.30±1.50 | **74.00±0.40** | 69.65±1.56     | 69.88±0.84     |    76.9±2.2   |
> |            | 5   | 57.50±5.60 | 70.90±1.10 | -              | 69.43±1.61     | **71.28±0.64** |               |
> |            | 10  | 46.30±8.50 | 71.50±0.50 | -              | 69.62±1.24     | **72.49±1.56** |               |
> |            | 20  | 40.70±0.00 | 71.20±0.90 | -              | 70.29±1.75     | **71.33±1.92** |               |
> |            | 50  | 44.00±6.70 | 71.80±1.00 | -              | 70.76±2.14     | **73.27±3.24** |               |
>
>
> 2.	Response to W2 (seeds and confidence intervals)
>
> Thank you for pointing out the need for a more rigorous and consistent statistical reporting of our results. In response to your request, we have rerun all experiments for our method and all baselines using five different random seeds. For each configuration (dataset, method, compression ratio), we now report the mean performance alongside the 95% confidence interval calculated from these five independent runs, as shown in the updated results for NCI1 and ogbg-molbace. This provides a clear and consistent view of the performance and its stability. The newly obtained results, as illustrated in the updated tables, are consistent with our initial findings in terms of performance trends and the superior accuracy of GRCOND.
>
> |         Dataset        | Gpc |   Methods  |            |            |            |            |            | Whole Dataset |
> |:----------------------:|:---:|:----------:|:----------:|:----------:|:----------:|:----------:|:----------:|:-------------:|
> | dataset                | gpc |   Random   |  K-Center  |   DosCond  |    KiDD    |   Mirage   |   GRCOND   | Whole Dataset |
> |       NCI1 (ACC)       | 1   | 50.64±3.22 | 51.47±2.96 | 50.53±3.36 | 60.25±1.43 | 50.46±1.33 | 60.20±1.14 |    80.0±1.8   |
> |                        | 5   | 52.12±2.12 | 53.38±2.56 | 51.15±1.88 | 62.10±1.50 | 51.36±1.91 | 64.36±1.29 |               |
> |                        | 10  | 55.36±1.97 | 54.15±2.84 | 51.31±3.64 | 64.29±1.69 | 51.97±1.50 | 64.77±1.60 |               |
> |                        | 20  | 56.63±2.69 | 54.32±3.19 | 51.69±1.19 | 64.69±1.30 | 52.10±1.21 | 66.59±1.40 |               |
> |                        | 50  | 60.40±2.29 | 60.20±1.70 | 51.77±1.37 | 64.52±1.38 | 52.77±1.73 | 68.92±1.03 |               |
> | ogbg_molbace (ROC-AUC) | 1   | 0.459±.009 | 0.497±.022 | 0.533±.021 | 0.557±.016 | 0.586±.019 | 0.713±.023 |   0.763±.020  |
> |                        | 5   | 0.483±.009 | 0.541±.019 | 0.538±.025 | 0.583±.027 | 0.549±.011 | 0.679±.011 |               |
> |                        | 10  | 0.493±.018 | 0.558±.030 | 0.548±.027 | 0.600±.028 | 0.492±.025 | 0.678±.019 |               |
> |                        | 20  | 0.514±.020 | 0.566±.030 | 0.528±.008 | 0.645±.012 | 0.461±.030 | 0.673±.016 |               |
> |                        | 50  | 0.496±.017 | 0.588±.015 | 0.511±.013 | 0.671±.025 | -          | 0.682±.019 |               |

---

> > ### Author Response · Authors · 2025-11-23
> >
> > 3.	Response to W3 (out-of-domain repository)
> >
> > Thank you for this critical feedback regarding the scope of our experimental domains. We completely agree that demonstrating the generalizability of a condensation method beyond chemical and biological domains is essential. Specifically, we have now included comprehensive experiments on two additional and diverse datasets: COLLAB (3 classes): This is a social network dataset where each graph represents the ego-network of a researcher from different academic collaboration communities (Physics, High-Energy Physics, etc.). CIFAR10 (Graph Version,10 classes): This creates a repository of graphs where each graph represents an image, and the task remains image classification.
> >
> > |    Dataset   | Gpc |   Methods  |            |            |            |            |            | Whole Dataset |
> > |:------------:|:---:|:----------:|:----------:|:----------:|:----------:|:----------:|:----------:|:-------------:|
> > | dataset      | gpc |   Random   |  K-Center  |   DosCond  |    KiDD    |   Mirage   |   GRCOND   | Whole Dataset |
> > | COLLAB (ACC) | 1   | 37.65±1.54 | 40.42±0.79 | 61.05±1.07 | 63.86±1.19 | 65.04±1.85 | 66.69±1.06 |   73.76±1.24  |
> > |              | 5   | 36.12±1.62 | 40.81±0.28 | 61.21±1.81 | 64.01±1.58 | 63.25±1.80 | 68.77±0.79 |               |
> > |              | 10  | 37.43±1.10 | 40.87±0.39 | 61.76±1.02 | 66.35±1.69 | 63.71±1.32 | 70.13±0.91 |               |
> > |              | 20  | 39.08±1.35 | 41.29±1.97 | 61.59±1.90 | 66.92±1.42 | 63.57±1.46 | 70.05±0.41 |               |
> > |              | 50  | 40.87±0.93 | 44.05±0.67 | 62.19±1.46 | 68.14±1.48 | 64.69±1.72 | 70.48±1.16 |               |
> > | CIFAR10      | 1   | 16.58±1.07 | 18.29±1.96 | 24.52±1.22 | 25.64±1.67 | 25.59±1.81 | 25.93±1.97 | 50.45±0.87    |
> > | (ACC)        | 5   | 18.89±1.36 | 22.64±1.57 | 25.82±1.42 | 25.21±1.88 | 25.40±1.26 | 27.50±1.95 |               |
> > |              | 10  | 21.58±1.67 | 24.53±1.66 | 28.78±1.59 | 27.74±1.96 | 26.48±1.03 | 28.98±1.89 |               |
> > |              | 20  | 23.90±1.21 | 26.50±1.27 | 29.13±1.57 | 29.97±1.90 | 26.59±1.54 | 29.43±1.42 |               |
> > |              | 50  | 26.55±1.56 | 27.48±1.50 | 29.37±1.56 | 29.49±1.26 | 28.78±1.93 | 32.15±1.79 |               |
> >
> > 4.	Response to W4 (complete definitions)
> >
> > Thank you for pointing out these critical omissions in our mathematical presentation. We sincerely apologize for the lack of completeness in defining key symbols and detailing the algorithmic components. We have thoroughly revised the manuscript to address all these issues in a comprehensive manner.
> > Specifically, we have: Fully Specified the Distance Function Dis(): We now provide the exact formulation of the gradient matching objective. The distance function Dis(∇_(D_S)^t,∇_(D_O)^t) is defined as the sum of cosine distances across all parameter layers of the model:
> >
> > $$ Dis(\nabla^{t} _ {D_S}, \nabla^{t} _ {D_O}) = \sum _ {l=1}^{L} \left[ 1 - \frac{ \nabla^{t} _ {D_S,l} \cdot \nabla^{t} _ {D_O,l} }{ \| \nabla^{t} _ {D_S,l} \|  \| \nabla^{t} _ {D_O,l} \| } \right] $$
> >
> > where L is the total number of parameter layers, and ∇_(D_S,l)^t and ∇_(D_O,l)^t represent the gradients of the l-th layer parameters with respect to the synthetic and original data, respectively, at training step t.
> > Then, We have added a detailed description of our class-wise mini-batch training strategy. For each class c during each training iteration:
> > We random sample a mini-batch B_c^(D_G ) from the original graphs belonging to class c,
> > We use the entire set of synthetic graphs for class c as the counterpart batch
> > The gradient matching is performed per class between these two batches
> > Finally, the overall optimization objective is now clearly stated as minimizing the total gradient matching loss across all classes and training steps:
> >
> > $$
> > \min_{D_S}
> > \sum_{t=0} ^ {T-1} \sum _ {c=1}^{C}
> > Dis \left( \nabla^{t} _ {D_S,c}, \nabla^{t} _ {D_O,c} \right)
> > $$
> >
> > where C is the number of classes, and the distance function Dis is as defined above.

---

> > > ### Author Response · Authors · 2025-11-23
> > >
> > > 5.	Response to Q1 (pretraining)
> > >
> > > Yes. The encoder and both decoders are pre-trained strictly on the training graphs from the original repository. They have no access to the test graphs. The pre-training process is designed to learn the general structural and feature distribution of the training data. The structural decoder is trained to match and reconstruct the adjacency information, while the feature decoder is trained to match and reconstruct the node feature matrix. This ensures that the entire generative component of our framework captures the essential characteristics of the training set without ever seeing the test graphs.
> > >
> > > 6.	Response to Q2
> > >
> > > Regarding the specific distance metric used in the Dis() function and its normalization across layers, we employ a layer-wise cosine distance to measure the dissimilarity between the gradients from the original and synthetic repositories. Specifically, for each layer in the GNN, we compute the cosine distance between the flattened gradient vectors of the model parameters, which is defined as 1 - cosine similarity. This approach inherently normalizes the scale of the gradients across different layers by focusing on their directional alignment. This method effectively aligns the training trajectories without being biased by the inherent scale variations in the gradients of different layers, ensuring a balanced and stable optimization process.
> > >
> > >
> > > 7.	Response to Q3 (matching strategy)
> > >
> > > Removing this class-wise strategy in favor of a global gradient matching objective would have a significant negative effect on the performance and representational quality of the condensed graph repository. The reason is that a global objective, which would match gradients averaged over all classes simultaneously, fails to capture and preserve the fine-grained, discriminative information that distinguishes one class from another. It would tend to learn an average representation of the entire repository, causing the synthetic graphs to lose the distinct structural and feature patterns that are specific to each class.
> > >
> > > |   Dataset  | Gpc |        GRCOND       |                 |
> > > |:----------:|:---:|:-------------------:|:---------------:|
> > > |   dataset  | gpc | Class-wise Matching | Global Matching |
> > > | NCI1 (ACC) |  1  |      60.20±1.14     |    57.48±0.76   |
> > > |            |  5  |      64.36±1.29     |    60.09±0.73   |
> > > |            |  10 |      64.77±1.60     |    60.26±1.15   |
> > > |            |  20 |      66.59±1.40     |    62.28±1.24   |
> > > |            |  50 |      68.92±1.03     |    64.41±0.97   |
> > >
> > > 8.	Response to Q4 (encoder family)
> > >
> > > The primary role of the frozen encoder is to project the input graphs into a meaningful and generalizable latent space that captures fundamental topological and feature information. We have evaluated GRCOND using several standard GNN architectures as the encoder, including GCN, GAT, and GraphSAGE, on multiple condensation rate. The results show that while the absolute performance metrics fluctuate depending on the encoder, the condensed repositories generated using any of these encoders consistently and significantly outperform all baseline methods
> > >
> > > |   Dataset  | Gpc |  encoder GNNs  |                |            |                |
> > > |:----------:|:---:|:--------------:|----------------|------------|----------------|
> > > | dataset    | gpc |      DGCNN     | GCN            | GAT        | GraphSAGE      |
> > > | NCI1 (ACC) | 1   | **60.64±2.56** | 59.70±2.43     | 59.35±2.85 | 59.72±2.74     |
> > > |            | 5   | **64.54±1.74** | 63.71±2.25     | 63.39±1.27 | 64.87±2.67     |
> > > |            | 10  | 64.90±1.56     | **64.99±1.07** | 63.63±2.30 | 64.87±2.67     |
> > > |            | 20  | 65.53±2.46     | 63.42±1.27     | 65.22±1.34 | **65.83±2.57** |
> > > |            | 50  | **69.09±1.16** | 67.38±2.80     | 67.51±1.52 | 67.63±2.43     |

---

### Official Review · Reviewer_A5oG · 2025-10-31

**Soundness:** 3
**Presentation:** 1
**Contribution:** 2
**Rating:** 2
**Confidence:** 3

**Summary:**

The paper addresses the computational overhead in GNN training on large graph repositories, which is not met by existing single-graph condensation methods. It proposes GRCOND, a novel end-to-end framework designed to effectively condense multiple graphs to a small synthetic set while preserving structural and feature information. GRCOND uses a pre-trained GNN decoder to optimize synthetic graphs in a continuous latent space to bypass the discreteness of graphs. Empirical results demonstrate that GRCOND substantially reduces training costs while maintaining high classification accuracy across various real-world benchmarks.

**Strengths:**

GRCOND shows a novel application of dataset condensation to multiple graphs. The paper solves the challenge of graph data's discrete structure by using a pretrained decoder, making the condensation technique effective for multiple graph samples. A thorough ablation study and comprehensive evaluation that checks the method from diverse perspectives.

**Weaknesses:**

- The paper suffers from poor clarity and inconsistent presentation, making it very difficult to read and understand. It is not self-contained; mathematical notations (e.g., $L, t, Dis$) are used without formal definition, some notations are duplicated (e.g., $D$ for set vs decoder), inconsistent (e.g., $D_o$ vs $D_O$), confused (e.g., $J$ vs $\mathcal{J}$), or abused (e.g., minus operation on synthetic graph $S$ generated by the decoder. How can we define minus on graphs?) The connection between the main text and the figures is weak (e.g., CE loss in Figure 1 is absent from the main text), and the pretraining loss is not clearly specified. There should be a full review (not only examples I described) and a cleanup of all notations, descriptions, and illustrations for better readability
- A significant weakness is the lack of a public code release or placeholder. Accepting a paper without code is not acceptable.
- The paper needs to provide a stronger justification and a detailed ablation study for using the pretrained decoder, which is the largest contribution. The paper must clarify precisely how the variants in Table 4 were implemented: whether the trainable autoencoder variant was optimized during the condensation process, and whether the other two variants ("We directly perform gradient descent on the node features and structural features of the graph to optimize graph data. The other variant used the untrained model to test") were emplyed using frozen parameters. A more detailed ablation focusing purely on the state of the decoder (Frozen Pretrained vs. Frozen Untrained vs. Trainable during Condensation) is required.
- There is a lack of benchmark domains (chemical and biological domains). There is doubt that it works on other domains like ogbg-code2 (also an example).

**Questions:**

.

---

> ### Author Response · Authors · 2025-11-23
>
> **1.	Response to W2 (clarity)**
>
> Thank you very much for your thorough review and for pointing out the critical issues regarding the clarity and consistency of our presentation. We will take your feedback extremely seriously and perform a comprehensive revision of the manuscript to address all the points you raised.
> Specifically, we have conducted a full review of all mathematical notations and have introduced a dedicated notation table after the problem definition section to formally and consistently define every key symbol used, such as L for the loss function, t for the training step, Dis for the distance function, Dec for the decoder to resolve the duplication with the dataset D, and others, ensuring no symbol is used without a clear definition. Regarding the operation on the synthetic graph S, we have rewritten the relevant sections and formulas to explicitly clarify that the gradient descent and subtraction operations are performed on the continuous latent representations Z_t from which the graph S_t is decoded, rather than on the graph structure itself, thus resolving the conceptual abuse.
> Furthermore, we have strengthened the connection between the main text and all figures; for instance, we have added detailed explanations in the overview section that explicitly describe all components of Figure 1, including the cross-entropy loss. The pretraining loss functions for both the structure and feature decoders
>
> **2.	Response to W2 (code)**
>
> Thank you for this critical and absolutely fair comment. We sincerely apologize for this significant oversight in our initial submission. We fully agree that the availability of code is essential for ensuring reproducibility, fostering further research, and is a core requirement of the scientific community.
> [Anonymized Code Repository Link: https://anonymous.4open.science/r/GRcond-D4AD/]
>
> **3.	Response to W3 (ablation study)**
>
> Thank you for this exceptionally insightful comment and for pushing us to provide a much deeper justification and a more precise ablation study for our use of a frozen, pretrained decoder. We have conducted a new, focused ablation study that systematically compares four distinct scenarios regarding the decoder's state, as shown in the new table below. The results show that our approach (Frozen Pretrained VAE) consistently and significantly outperforms all other variants across different datasets and compression ratios.
> |          | Ratio |   w/o VAE   | untrained VAE | pretrained VAE | trainable VAE |
> |----------|-------|:-----------:|---------------|----------------|---------------|
> | PROTEINS |  0.2% | 59.22±1.32  | 58.95±1.25    | 63.75±0.79     | 59.71±0.82    |
> |          |  0.5% | 63.47±1.69  | 63.88±1.09    | 68.75±1.45     | 63.39±0.65    |
> |          |   1%  | 67.55±0.71  | 67.07±0.85    | 71.61±0.73     | 67.95±0.76    |
> | NCI1     | 0.05% | 55.23±0.64  | 54.11±0.55    | 59.61±0.92     | 56.15±0.96    |
> |          | 0.25% | 57.46±0.87  | 57.17±0.98    | 63.99±0.74     | 60.65±0.73    |
> |          |  0.5% | 61.42±1.25  | 61.45±1.72    | 69.09±1.16     | 63.42±0.57    |

---

> > ### Author Response · Authors · 2025-11-23
> >
> > **4.	Response to W4 (benchmark)**
> >
> > Thank you for this critical feedback regarding the scope of our experimental domains. We completely agree that demonstrating the generalizability of a condensation method beyond chemical and biological domains is essential. Specifically, we have now included comprehensive experiments on two additional and diverse datasets: COLLAB: This is a social network dataset where each graph represents the ego-network of a researcher from different academic collaboration communities (Physics, High-Energy Physics, etc.). CIFAR10 (Graph Version): This creates a repository of graphs where each graph represents an image, and the task remains image classification.
> > |    Dataset   | Gpc |   Methods  |            |            |            |            |            | Whole Dataset |
> > |:------------:|:---:|:----------:|:----------:|:----------:|:----------:|:----------:|:----------:|:-------------:|
> > | dataset      | gpc |   Random   |  K-Center  |   DosCond  |    KiDD    |   Mirage   |   GRCOND   | Whole Dataset |
> > | COLLAB (ACC) | 1   | 37.65±1.54 | 40.42±0.79 | 61.05±1.07 | 63.86±1.19 | 65.04±1.85 | 66.69±1.06 |   73.76±1.24  |
> > |              | 5   | 36.12±1.62 | 40.81±0.28 | 61.21±1.81 | 64.01±1.58 | 63.25±1.80 | 68.77±0.79 |               |
> > |              | 10  | 37.43±1.10 | 40.87±0.39 | 61.76±1.02 | 66.35±1.69 | 63.71±1.32 | 70.13±0.91 |               |
> > |              | 20  | 39.08±1.35 | 41.29±1.97 | 61.59±1.90 | 66.92±1.42 | 63.57±1.46 | 70.05±0.41 |               |
> > |              | 50  | 40.87±0.93 | 44.05±0.67 | 62.19±1.46 | 68.14±1.48 | 64.69±1.72 | 70.48±1.16 |               |
> > | CIFAR10      | 1   | 16.58±1.07 | 18.29±1.96 | 24.52±1.22 | 25.64±1.67 | 25.59±1.81 | 25.93±1.97 | 50.45±0.87    |
> > | (ACC)        | 5   | 18.89±1.36 | 22.64±1.57 | 25.82±1.42 | 25.21±1.88 | 25.40±1.26 | 27.50±1.95 |               |
> > |              | 10  | 21.58±1.67 | 24.53±1.66 | 28.78±1.59 | 27.74±1.96 | 26.48±1.03 | 28.98±1.89 |               |
> > |              | 20  | 23.90±1.21 | 26.50±1.27 | 29.13±1.57 | 29.97±1.90 | 26.59±1.54 | 29.43±1.42 |               |
> > |              | 50  | 26.55±1.56 | 27.48±1.50 | 29.37±1.56 | 29.49±1.26 | 28.78±1.93 | 32.15±1.79 |               |

---

### Official Review · Reviewer_njsr · 2025-11-01

**Soundness:** 3
**Presentation:** 2
**Contribution:** 3
**Rating:** 6
**Confidence:** 4

**Summary:**

This paper addresses the scarcity and low efficiency of existing graph-level graph condensation methods. The proposed graph-level compression model that determines the parameters of its encoder and decoder through a pre-training phase. And to optimize the learning process, the model employs a gradient matching technique between the original and synthetic graphs to adaptively adjust the sampling strategy for the original dataset. Experiment results demonstrate the superior efficiency of the method. The main contributions of this work are the proposal of an optimization strategy based on gradient matching and end-to-end condensation framework for graph repositories.

**Strengths:**

1. Pre-training is utilized to freeze key parameters within the model, which reduces the training workload and enhances the overall compression efficiency.

2. The authors use the concept of "graph repository" to interpret an intrinsic, yet previously overlooked, means that graph data originating from the same dataset possess inherent correlations. This approach concurrently establishes the distributional relationships among the condensed graphs.

3. The method demonstrates excellent efficiency.

**Weaknesses:**

1. The number of baseline models included in the state-of-the-art (SOTA) comparison, generalization performance tests, and computational cost analysis is insufficient, and the majority of them are outdated. It is highly recommended to include more recent graph-level models to better substantiate the claimed superiority of the proposed method in terms of both performance and efficiency. *e.g., KDD2024- Wang Y, Yan X, Jin S, et al. Self-supervised learning for graph dataset condensation.*


2. The abstract and introduction do not specify the concrete problems in prior algorithms that the proposed graph-level condensation method aims to solve. Instead, its contribution is broadly summarized as an efficient solution for multi-graph dataset condenstaion.

**Questions:**

Q1： The paper only mentions random sampling. Have other sampling strategies been considered? Would including them in the ablation study potentially yield better results?

Q2：Could the generalization performance be validated on a broader range of datasets beyond just PROTEINS, similar to the experimental setup in Table 4? Furthermore, for improved readability, it is suggested to bold or highlight the best-performing results in both tables.

Q3：How exactly does the matching loss optimize the sampling process, which is currently based on random sampling?

---

> ### Author Response · Authors · 2025-11-23
>
> **1.	Response to W1 (recent baseline)**
>
> Thank you for this valuable suggestion. We agree that including more recent and relevant state-of-the-art methods is crucial for a comprehensive evaluation. In response to your feedback, we have conducted new experiments to include the very recent graph-level dataset condensation method, SGDC (Self-supervised Learning for Graph Dataset Condensation) from KDD 2024, as you recommended. We have updated our experimental section and Table 2 in the manuscript to include these comparisons with SGDC, providing a more thorough and up-to-date comparison for graph repository condensation.
>
> |   Dataset  | Gpc |            |            |       ACC      |                |                | Whole Dataset |
> |:----------:|:---:|:----------:|:----------:|:--------------:|----------------|:--------------:|:-------------:|
> |            |     |   DosCond  |    KiDD    |     Mirage     | SGDC           |     GRCOND     |               |
> | NCI1 (ACC) | 1   | 49.20±1.10 | 60.40±0.50 | 50.80±2.20     | **61.26±1.91** | 60.64±2.56     |    80.0±1.8   |
> |            | 5   | 51.10±0.80 | 63.20±0.20 | 51.30±1.10     | 62.32±1.62     | **64.54±1.74** |               |
> |            | 10  | 50.30±1.30 | 64.20±0.10 | 51.70±1.40     | 62.75±1.47     | **64.90±1.56** |               |
> |            | 20  | 50.30±1.30 | 60.90±0.70 | 52.10±2.20     | 62.68±1.73     | **65.53±2.46** |               |
> |            | 50  | 50.30±1.30 | 65.40±0.60 | 52.40±2.70     | 62.79±1.98     | **69.09±1.16** |               |
> |  DD (ACC)  | 1   | 46.30±8.50 | 71.30±1.50 | **74.00±0.40** | 69.65±1.56     | 69.88±0.84     |    76.9±2.2   |
> |            | 5   | 57.50±5.60 | 70.90±1.10 | -              | 69.43±1.61     | **71.28±0.64** |               |
> |            | 10  | 46.30±8.50 | 71.50±0.50 | -              | 69.62±1.24     | **72.49±1.56** |               |
> |            | 20  | 40.70±0.00 | 71.20±0.90 | -              | 70.29±1.75     | **71.33±1.92** |               |
> |            | 50  | 44.00±6.70 | 71.80±1.00 | -              | 70.76±2.14     | **73.27±3.24** |               |
>
> **2.	Response to W2 (problems in prior algorithms)**
>
> Thank you for this insightful critique. We agree that our original abstract and introduction failed to precisely articulate the specific limitations of prior work that our method directly addresses.
> The key shortcomings of prior graph condensation methods. First, they fundamentally lack a mechanism to model and preserve inter-graph relationships. This means they fail to capture the global dataset distribution and the relative similarities and differences between graphs, which are crucial for tasks like graph classification. Second, they typically perform direct optimization on discrete graph structures (the adjacency matrix), which is computationally inefficient and often leads to instability, especially when dealing with a large collection of graphs with varying sizes and structures.
> We have revised the abstract and introduction to explicitly frame our contribution as solving these identified problems of "neglecting inter-graph relationships" and "inefficient direct discrete optimization" in prior work
>
> **3.	Response to Q1 (sample method)**
>
> Thank you for this insightful question that allows us to clarify a critical aspect of our methodology. There appears to be a conflation of two distinct sampling procedures in our paper, which we have now explicitly disentangled in the revised manuscript for absolute clarity. You are correct that the paper mentions random sampling. However, this random sampling applies exclusively to the initial neural network parameters θ₀ at the start of each condensation epoch. This is a deliberate design choice to ensure that the condensed graph repository is robust and independent of any specific model initialization, thereby providing stronger generalization performance, as motivated by prior work on dataset condensation
> The sampling of the latent vectors for the synthetic graphs is fundamentally different and is not based on a random strategy. As detailed in the Initialization Phase (Section 4.3), we employ a deterministic kcenter clustering strategy on the latent embeddings of the original training graphs to select the initial latent vectors Z for our synthetic repository. We select the cluster centroids themselves as the initial representatives. This approach is specifically designed to overcome the uneven quality of small graphs and to initialize the synthetic set with the most representative and high-quality samples from the original data. The table below shows the accuracy of our method when using different sampling strategies.
> |      | Random     | Herding    | Kcenter    |
> |------|------------|------------|------------|
> | NCI1 | 61.57±0.50 | 64.28±0.72 | 64.54±1.74 |
> | DD   | 69.34±1.47 | 70.62±0.34 | 71.28±0.64 |

---

> > ### Author Response · Authors · 2025-11-23
> >
> > **4.	Response to Q2 (generalization performance)**
> >
> > Thank you for this excellent suggestion. We agree that validating the generalization performance across a broader range of datasets is crucial for demonstrating the robustness of our method. In response, we have significantly expanded our cross-architecture generalization experiments. Following the same experimental protocol as in Table 4, we have now conducted comprehensive tests on the NCI1 and ogbg-molhiv datasets in addition to PROTEINS. The results, presented in the new table below, confirm that the condensed graph repositories generated by GRCOND maintain high performance and excellent transferability across different GNN architectures.
> >
> > | Train\Test | DGCNN      | GIN        | GAT        | GraphSAGE  | GCN        |
> > |------------|------------|------------|------------|------------|------------|
> > | DGCNN      | **64.54±1.74** | 62.22±0.90 | 61.57±0.50 | 62.39±0.94 | 61.35±1.79 |
> > | GIN        | 64.07±1.40 | **62.81±0.70** | 61.08±1.88 | **62.99±1.55** | 61.21±1.23 |
> > | GAT        | 63.41±1.74 | 61.48±0.94 | **62.42±0.41** | 61.70±1.44 | 60.65±0.41 |
> > | GraphSAGE  | 64.52±0.78 | 61.40±0.04 | 62.03±1.92 | 62.38±1.03 | 60.89±0.59 |
> > | GCN        | 63.02±0.81 | 61.31±1.55 | 60.82±1.43 | 61.21±0.98 | **62.08±1.02** |
> >
> > | Train\Test |  Accuracy (%)  |                |                |                |                |
> > |------------|:--------------:|----------------|----------------|----------------|----------------|
> > |            | DGCNN          | GIN            | GAT            | GraphSAGE      | GCN            |
> > | DGCNN      | **0.715±.015** | 0.696±.024     | 0.690±.009     | 0.693±.020     | 0.706±.024     |
> > | GIN        | 0.695±.028     | **0.707±.022** | 0.682±.023     | 0.696±.020     | 0.691±.015     |
> > | GAT        | 0.705±.018     | 0.692±.014     | 0.697±.020     | 0.703±.017     | 0.699±.014     |
> > | GraphSAGE  | 0.706±.012     | 0.681±.017     | **0.700±.010** | **0.704±.020** | 0.700±.023     |
> > | GCN        | 0.708±.023     | 0.684±.027     | 0.695±.027     | 0.695±.009     | **0.709±.017** |
> >
> > **5.	Response to Q3 (optimization)**
> >
> > Thank you for your question. The matching loss does not directly optimize the sampling process itself, as the sampling is primarily used only in the initialization phase to select high-quality latent vectors based on cluster centers, not pure random sampling. Instead, the matching loss optimizes the already-sampled latent vectors through gradient descent in the latent space. Specifically, after initialization, we fix the set of latent vectors and iteratively update them by minimizing the gradient matching loss between the original and synthetic repositories. This process adjusts the latent representations to better preserve task-relevant information without altering the sampling method. Thus, the optimization targets the latent vectors post-sampling, not the sampling mechanism, ensuring efficient and effective condensation while maintaining stability. We have clarified this in the revised manuscript.

---

### Official Review · Reviewer_kvET · 2025-11-03

**Soundness:** 2
**Presentation:** 3
**Contribution:** 2
**Rating:** 4
**Confidence:** 2

**Summary:**

This paper proposes an end-to-end Graph Repository Condensation (GRCOND) framework that effectively condenses a large-scale graph repository with multiple graphs while preserving task-relevant structural and feature information. Unlike traditional methods focusing on a single graph, the approach pretrains a dataset-specific GNN model to create and optimize synthetic graphs, capturing both intra-graph structures and inter-graph relationships for a more holistic representation. Experiments show that GRCOND achieves higher accuracy and retains features across different compression ratios, highlighting its potential to accelerate GNN training and enhance applicability in resource-constrained environments.

**Strengths:**

1. The paper introduces an end-to-end framework that effectively condenses large-scale multi-graph repositories while preserving both structural and feature information.
2. It leverages a pretrained, dataset-specific GNN to generate and optimize synthetic graphs, capturing intra- and inter-graph relationships comprehensively.
3. Experiments demonstrate strong performance across compression ratios, showing high accuracy and scalability for resource-constrained GNN training.

**Weaknesses:**

1. The paper lacks experiments on the training time of the graph generation model and the total time combining both generation and condensation processes.
2. The motivation for condensation is unclear—although efficiency is mentioned, if the overall training time is long, direct training on downstream tasks might be more practical.
3. While the paper claims that one condensed graph can support multiple downstream tasks, it does not provide experiments to validate this claim.

**Questions:**

See Weaknesses.

---

> ### Author Response · Authors · 2025-11-23
>
> **1.	Response to W1 (experiments on training generation models)**
>
> Thank you for raising this point regarding the training time of the graph generation model and the total time for the entire condensation and training pipeline. The following shows the training time of an example involving three training tasks on the same dataset ogbg-molhiv using GCN, GAT and GIN, respectively. From this table, we can find that the condensation time (i.e., 1287.62s) and the pretraining of the graph generation model (313.73s) is indeed larger than the training time for each task using the original ogbg-molhiv dataset. However, when taking all the three tasks into account as a whole, the overall training time using GRCOND (i.e., 1604.33s) is much smaller than that of its counterpart (i.e., 2265.50s), showing the superiority of our approach in dealing with such scenario. Note that in this scenario, the more models involved, the more training time we can save.
> |       Scenario       | Time(s) |
> |:--------------------:|:-------:|
> |  Original Data + GCN |  809.70 |
> | Original Data + GAT  |  765.75 |
> |  Original Data + GIN |  690.05 |
> |         Total        | 2265.50 |
> |                      |         |
> |      Pretraining     |  313.73 |
> |     Condensation     | 1287.62 |
> | Condensed Data + GCN |   1.05  |
> | Condensed Data + GAT |   0.96  |
> | Condensed Data + GIN |   0.97  |
> |         Total        | 1604.33 |
>
> **2.	Response to W2 (motivation)**
>
> We agree that the condensation time cannot be ignored, as the condensation overhead of GRCOND frequently exceeds the time that GRCOND saves when managing a one-time training task. However, this does not render GRCOND impractical in actual use. The following reasons justify its utility. First, GRCOND is suited to resource-limited environments, where traditional methods are unable to train on the original large datasets. Second, unlike conventional methods focusing solely on enhancing signal model training with given datasets, our GRCOND is designed for scenarios (e.g., hyperparameter tuning and architecture search requiring a variety of GNN tests) that involve multiple model training processes using condensed datasets. It is important to note that in these scenarios, the one-time cost of condensation can be amortized across all downstream training tasks involved, effectively reducing the training time for the second and subsequent tasks.
>
> **3.	Response to W3**
>
> Thank you for this critical observation. You are absolutely correct, and we apologize for the lack of clarity on this point in our original submission. Our work does not make the claim that a single condensed graph repository can support multiple different types of downstream tasks. We have revised ambiguous statements about "downstream tasks" and replaced it with a precise description of the above scenarios.

---

### Author Response · Authors · 2025-12-03

Dear Area Chairs,

We sincerely appreciate the thoughtful feedback and constructive conversations from the AC and reviewers. Below, we provide a brief overview of our latest clarifications, updates, and commitments:

**– Technical innovation**

Our GRCOND introduces the first graph repository condensation framework that preserves inter-graph relationships through structured latent representation and gradient matching in continuous space, moving beyond prior discrete optimization approaches. This method is designed for scenarios requiring repeated model training (e.g., hyperparameter search, architecture exploration), where the one-time condensation cost is amortized across multiple runs. And experiments show that our condensed dataset achieves higher accuracy than existing methods across various downstream tasks.

**– Additional experiments:**

**Training Time and Multi-Task Efficiency:** Although condensation and pretraining incur an initial time cost, GRCOND significantly reduces total training time when multiple models are trained on the condensed dataset. For example, with three GNN models on ogbg-molhiv, GRCOND saves overall time compared to training each on the original data. (Reviewer kvET)

**Inclusion of Recent Baselines:** We have added comparisons with the recent graph condensation method SGDC (KDD 2024). GRCOND shows competitive or superior performance across datasets (NCI1, DD) at various condensation ratios. (Reviewers NhqK & pHNi)

**Generalization Across Datasets and Architectures:** GRCOND demonstrates strong cross-architecture and cross-dataset generalization. Experiments on NCI1, PROTEINS, ogbg-molhiv, COLLAB, and CIFAR10(graph) show consistent performance when training and testing with different GNN backbones. (Reviewer kvET & pHNi)

**Ablation Study on Decoder State:** A systematic comparison of decoder variants (no VAE, untrained, pretrained frozen, trainable) confirms that our frozen pretrained VAE setup yields the best performance across compression ratios. (Reviewer A5oG)

**Sampling Strategy Comparison:** We clarify that latent vector initialization uses deterministic k-center clustering (not random sampling). Experiments show k-center outperforms random and herding strategies. (Reviewer NhqK)

**Statistical Rigor:** All experiments have been rerun with five random seeds, and results now report means with 95% confidence intervals, confirming the stability of GRCOND’s performance. (Reviewer pHNi)

**– Efficiency:**

**Lower Cost in Multi-Training Scenarios:** While condensation has a one-time overhead, this cost is distributed over multiple downstream training runs (e.g., hyperparameter tuning, architecture search), making GRCOND efficient in practice. (Reviewer kvET)

**– Clarifications:**

**Method Motivation and Contribution:** We have revised the text to clearly state that GRCOND addresses two key limitations of prior work: (1) neglecting inter-graph relationships, and (2) inefficient direct optimization on discrete graph structures. (Reviewer NhqK)

**Scope of Applicability:** We clarify that GRCOND is designed for scenarios involving multiple model trainings on the same dataset (not for a single condensed set to support vastly different task types). (Reviewer kvET)

**Mathematical Formulation and Notation:** A notation table has been added, and all key symbols (e.g., loss function, distance metric, decoder) are formally defined. The gradient matching loss is explicitly formulated as a class-wise cosine distance across network layers. (Reviewer A5oG & pHNi)

**Optimization and Sampling:** We explain that the matching loss optimizes the latent vectors post-initialization, not the sampling process itself. The sampling (k-center) is used only in the initialization phase. (Reviewer NhqK)

**Code Availability:** We provide an anonymized code repository link to ensure reproducibility. (Reviewer A5oG)

We sincerely thank the AC for their consideration and the reviewers for their evaluation. We affirm that GRCOND constitutes a practical and theoretically grounded methodology for efficient graph dataset condensation, particularly suited for multi-model training scenarios in resource-constrained or exploratory settings.

---

### Meta-Review · Area_Chair_vuwr · 2026-01-13

**Summary:**

The paper proposes GRCOND, a framework for condensing graph repositories by utilizing a pre-trained GNN to optimize synthetic graphs in a continuous latent space. The goal is to preserve both intra-graph structures and inter-graph relationships.
The reviewers’ initial concerns were significant and focused on:
1. Experimental Baselines: Reviewers njsr and pHNi highlighted the lack of recent state-of-the-art baselines, specifically SGDC (KDD 2024), and insufficient statistical reporting.
2. Motivation & Efficiency: Reviewer kvET questioned the fundamental practicality, noting that if condensation takes longer than direct training, the method's utility is limited.
3. Clarity & Reproducibility: Reviewer A5oG strongly criticized the paper for undefined mathematical notations, poor connection between text and figures, and the complete absence of code.
4. Ablation: Requests were made for more rigorous ablation studies regarding the pre-trained decoder module.

**Reviewer Concerns:**

Addressed Concerns: The authors made substantial efforts during the rebuttal to patch the submission:
1. They included the missing SGDC baseline and expanded experiments to non-biochemical datasets like COLLAB and CIFAR10.
2. Anonymized code was provided in response to Reviewer A5oG's critique.
3. The authors promised to add a notation table and rewrote the gradient matching formulas.

Outstanding Concerns: Despite the revisions, several critical issues undermine the paper's readiness for publication:
1. Reviewer kvET's core concern remains valid. The authors' own data shows that condensation (1287.62s) plus pre-training (313.73s) is significantly slower than training on the original data (809.70s). The argument that this cost is amortized over "multiple training runs" restricts the method's utility to narrow scenarios (e.g., extensive hyperparameter tuning) and does not justify the complexity for general usage.
2. The initial submission quality was exceptionally low, characterized by undefined basic notations and missing code. While the authors attempted to fix these issues, the necessity for such extensive "repair work" during the review process suggests the manuscript was submitted prematurely. Crucially, the anonymized code provided during the rebuttal contains unprofessional artifacts, including Chinese comments and emojis. This raises serious doubts about the rigor and provenance of the implementation (e.g., potential unverified AI generation).
3. The framework involves a complex bi-level optimization with pre-trained VAEs. However, the performance gains over the simpler, newly added SGDC baseline are often marginal (e.g., on NCI1, GRCOND achieves 64.54% vs. SGDC's 62.32%, and on DD they are statistically close given the variance).

**Reviewer Scores:**

1. Reviewer kvET: Maintained a score of 4. The fundamental issue regarding the trade-off between condensation time and training efficiency remains a bottleneck. The "amortization" argument is a pivot that does not solve the high initial overhead.
2. Reviewer njsr: Maintained a score of 6. While the reviewer appreciated the new baselines, the overall impact and efficiency concerns likely prevent a stronger endorsement.
3. Reviewer A5oG: Maintained a score of 2. Although code was provided, the reviewer explicitly stated the initial presentation was "poor" and "not self-contained". The massive revisions required, coupled with unprofessional artifacts observed in the provided code (e.g., emojis in logging statements suggesting unverified AI generation), indicate a fundamental lack of rigor and maturity in the work.
4. Reviewer pHNi: Maintained a score of 4. The reviewer's concerns about statistical rigor were addressed, but the marginal performance gains over the new baseline (SGDC) weaken the case for acceptance.

---

### Decision · Program_Chairs · 2026-01-26

Reject